# Cancer and mortality risks among people with multiple sclerosis: A population-based study in Isfahan, Iran

Amirhossein Nafari[1‡], Saeed Vaheb[1‡], Alireza Afshari-Safavi[1,2], Zahra Ravankhah[3], Fotooheh Teimouri[3], Vahid Shaygannejad [1,4]*, Omid Mirmosayyeb[1,4]

1 Isfahan Neurosciences Research Center, Isfahan University of Medical Sciences, Isfahan, Iran,
2 Department of Biostatistics and Epidemiology, Faculty of Health, North Khorasan University of Medical Sciences, Bojnurd, Iran, 3 Department of Noncommunicable Disease, Faculty University of Medical Sciences, Isfahan, Iran, 4 Department of Neurology, School of Medicine, Isfahan University of Medical Sciences, Isfahan, Iran

‡ AN and SV contributed equally to this work as first authors.
* v.shaygannejad@gmail.com

## Abstract

### Background

Multiple sclerosis (MS) and cancer present substantial global health challenges. Understanding cancer patterns among people with MS (PwMS) is crucial due to potential variations across demographics and geographic regions. Isfahan province in Iran, known for its high MS incidence ratio, offers a significant population for comprehensive studies on MS. In this study, we aim to investigate the association between risk of cancer and MS.

### Method

Data on PwMS were collected utilizing the National Multiple Sclerosis Registry System of Iran (NMSRI), with diagnoses confirmed using McDonald criteria by neurologists specialized in MS. Cancer incidence was investigated using the Iranian National Population-Based Cancer Registry (INPCR) data, collected following international protocols. Descriptive statistics and regression analyses were employed to assess factors associated with cancer and mortality risks among PwMS. Survival analysis was conducted using Kaplan-Meier curves.

### Results

Out of 10,049 PwMS, 123 were diagnosed with cancer, with an mean age at the time of cancer diagnosis being 40.41 years and a mean MS duration of 6.76 years. The majority had relapsing-remitting MS (81.2%), and Interferon-β was the most common disease-modifying therapy (DMT) (42.4%). Cancer incidence was 125.6 per 100,000 person-years, peaking at ages 60–64 (677.9 per 100,000 person-years). Receiving monoclonal antibody medications and older age were significantly associated with higher cancer risk (OR:1.542 (1.009–2.357), OR:1.033 (1.015–1.051), respectively). Female breast cancer had the highest incidence ratio among PwMS (40.17 per 100,000 person-years), followed by thyroid (18.38 per 100,000 *person-years*) and digestive system cancers (17.36 per 100,000 *person-years*).

**Data Availability Statement:** The data used in this study belongs to the MUI, and we had access to

certain data for research purposes. Anonymized data will be available upon reasonable request from any qualified investigator by contacting the corresponding author or the Isfahan Neurosciences Research Center at MUI via email at neuroscience@mui.ac.ir, in compliance with the General Data Protection Regulation. To ensure long-term storage and availability, the data will be securely archived and stored according to institutional data retention policies.

**Funding:** The author(s) received no specific funding for this work.

**Competing interests:** NO authors have competing interests.

Breast cancer was the predominant cancer in women, while digestive system cancers were most common among men. Being male and having longer MS duration were linked to higher cancer mortality risk (HR: 2.683, 1.087, respectively).

## Conclusion

Cancer incidence among 10,049 people with multiple sclerosis was significant, especially in older individuals, with breast cancer being the most common. Male gender and longer MS duration were linked to higher cancer mortality risk.

## Introduction

Multiple sclerosis (MS) is the most common chronic inflammatory and neurodegenerative disorder characterized by demyelination of the central nervous system, resulting in a broad spectrum of neurological impairments [1, 2]. With a global prevalence of approximately 2.8 million individuals, MS is also on the rise in developing countries and among children [3]. Over the course of MS, patients are exposed to various opportunities for developing additional illnesses, including cancer [4].

The coexistence of MS and cancer has intrigued the scientific community due to conflicting findings regarding the potential risk of cancer among people with MS (PwMS). While some studies suggest a decreased susceptibility to cancer in PwMS, others indicate an increased or comparable risk [5, 6].

It is crucial to comprehend the potential mechanisms underlying the coincidence of MS and cancer in order to establish a possible connection between them. Early studies of the MS-cancer association hypothesized that the immune dysregulation associated with MS could confer a protective effect against cancer development [7]. However, chronic inflammation and the use of specific immunosuppressive treatments could potentially weaken the immune system's defense against cancer or make the immune system pro-tumorigenic [8].

Understanding these mechanisms requires consideration of immunomodulatory treatments frequently employed to manage MS symptoms. There is contradictory information regarding the potential impact of long-term exposure to Disease-Modifying Therapies (DMTs) on the risk of developing neoplasms [9]. While a number of studies indicate that Glatiramer Acetate and Interferon-β do not raise the overall risk of cancer, it has been proposed that immunosuppressive drugs such as cladribine, fingolimod, natalizumab, and alemtuzumab may carry a potential cancer risk due to their action on the immune system and the lack of long-term data [6]. However, there is a lack of definitive data on the actual risk associated with DMT exposure, and factors contributing to cancer risk are often overlooked in these studies [9].

Epidemiological data contributes vital insights into the observed patterns of cancer occurrence within the MS population. Variations across demographics and geographic regions further highlight the need for a comprehensive evaluation of cancer risk in different MS populations. While the overall likelihood of developing cancer may not surpass that of the general population, PwMS might face lower survival rates [10]. The most common cancers among PwMS are cervical, breast, and digestive cancers [11].

Isfahan has consistently exhibited one of the highest MS prevalence rates in Iran [12, 13], with the prevalence reaching 183.9 per 100,000 in 2021, marking it as one of the highest rates in the Middle East and Asia [12]. This study aims to evaluate the prevalence and incidence of cancer among PwMS in Isfahan, considering the intriguing association between MS and

cancer. Epidemiological data is essential for understanding cancer patterns in the diverse MS population, especially considering variations across demographics and geographic regions. The main goal of this study is to illuminate the cancer risks within the PwMS to inform future clinical practices and research.

## Methods

### Study population

This retrospective population-based study aimed to evaluate the incidence and prevalence of cancer among PwMS in Isfahan province, Iran, utilizing data from the National Multiple Sclerosis Registry System of Iran (NMSRI)and Iranian National Population-Based Cancer Registry (INPCR). Data were accessed for research purposes from September 2022 to September 2023. Iran, located in western Asia, is a vast country comprising 423 counties and 31 provinces, covering a total area of 1,648,195 square kilometers and with a population of approximately 80 million, according to the last national census in 2016 [14]. Isfahan, situated in central Iran(31˚ 26′–34˚30′ N, 49˚30′–55˚50′ E), is one of the most populous and largest provinces, with a population of 5,386,437 residents and covering an area of 107,000 square kilometers [15].

### Data collection methodology for people with multiple sclerosis

Data on the incidence of MS in the Isfahan population were obtained from the NMSRI [16]. Detailed information regarding the data collection method for PwMS has been provided in our previous research [12] Patients diagnosed with MS in Isfahan between 1996 and 2022 were referred to the scientific committee of medical university of Isfahan (MUI), which consists of neurologists specializing in MS and central nervous system demyelinating disorders, for confirmation of their diagnosis using the McDonald criteria [17]. Once approved, these patients were recorded in the database, and relevant information—including birth date, sex, MS onset date, course of MS, and DMT was collected for analysis.

### Data collection methodology for cancer patients

Cancer incidence among PwMS was investigated using data from the INPCR [16, 18]. The INPCR was established by the Ministry of Health and Medical Education in the early 2010s with the aim of developing comprehensive guidelines for population-based cancer registries and collecting standardized cancer data. Data collection commenced in 2014, following international protocols recommended by organizations such as the International Agency for Research on Cancer and the International Association of Cancer Registries. The INPCR utilized pathology reports, clinical histories, and death certificates from various sources, including hospitals and pathology labs, and employed the International Classification of Diseases for Oncology (ICD-O-3) for coding tumor characteristics [19]. Quality control measures were implemented to ensure data accuracy and completeness.

### Inclusion and exclusion criteria

In this retrospective population-based study, inclusion criteria comprised individuals residing in Isfahan province, Iran, who were diagnosed with MS according to the McDonald criteria and confirmed by the scientific committee of the MUI. Additionally, cancer diagnoses had to be confirmed using data from the INPCR from 2014 to 2023, through pathology reports, clinical histories, or death certificates.

Furthermore, individuals who could not be identified through their national code, those who did not participate in follow-up interviews, those with insufficient data, or those whose cancer was diagnosed prior to the onset of MS were excluded from the analysis.

Individuals diagnosed with both cancer and MS were identified via their national code. Surviving patients were asked to join interviews where their clinical characteristics, family medical history, diagnostic examinations, treatment details, disease status, and overall outcomes were documented. Disability assessment for PwMS was conducted using the Expanded Disability Status Scale (EDSS) [20], and the clinical course of MS was categorized as relapsing-remitting MS (RRMS), secondary progressive MS (SPMS), or primary progressive MS (PPMS) [21]. These clinical parameters were assessed through clinical examinations conducted during the interview.

## Ethical approval and compliance

Ethical approval for the present study was obtained from the ethical committee of Isfahan University of Medical Sciences (Ethical code: IR.MUI.MED.REC.1400.268), and the study was conducted in compliance with the principles outlined in the Declaration of Helsinki and its subsequent revisions.

## Statistical analysis

Descriptive statistics were employed to present data, with categorical variables reported as frequency (percentage), normal continuous variables as mean and standard deviation (SD), and non-normal continuous variables as median (interquartile range). The relationship between various factors—such as age, age at diagnosis, sex, MS type, education level, and DMT—and the likelihood of developing cancer in PwMS was assessed using both univariable and multivariable logistic regression analyses. Additionally, Cox regression was employed to evaluate the risk of death in MS-cancer patients. Survival curves were generated using the Kaplan-Meier method, stratified by sex and MS type. Cancer incidence ratio were calculated overall, by sex, and across age at MS subgroups, representing the number of newly confirmed cases per 100,000 population divided by the sum of person-time (years) at risk. The person-time at risk was determined from the date of MS diagnosis until the onset age of cancer diagnosis or the end of follow-up. Propensity score matching was employed as a weighting method to mitigate the impact of confounding variables. All statistical analyses were conducted using SPSS software (version 20, IBM Corporation, Armonk, NY).

## Results

### Demographic characteristics of people with multiple sclerosis

In the baseline sample of 10,049 participants diagnosed with MS in Isfahan, the mean age of patients was 41.68 years (SD = 10.34), with a mean age at MS diagnosis of 31.76 years (SD = 10.08). Most PwMS had RRMS (86.2%), followed by SPMS (12.7%) and PPMS (1.1%). The most common DMT among PwMS was Interferon-β (42.4%), followed by Monoclonal antibodies (20.7%) and Dimethyl fumarate (11%). Among these participants, 126 individuals were diagnosed with both MS and cancer. Among MS-cancer patients, the most common DMT was Interferon-β (41.5%), followed by Monoclonal antibodies (30.9%) and Teriflunomide (8.1%) (Table 1).

### Incidence ratio of cancer among people with multiple sclerosis

The overall incidence ratio of cancer among PwMS was 125.6 (95% CI: 104.4–149.9) per 100,000 person-years, with 123 cases identified among 10,049 PwMS (S1 Table). See study

**Table 1. Demographics and characteristics of people with multiple sclerosis with and without cancer.**

| Parameter | | Overall | MS without cancer (n = 9926) | MS-cancer (n = 123) |
|---|---|---|---|---|
| Age; mean (SD) | | 41.68 (10.34) | 41.63 (10.32) | 45.93 (11.58) |
| Age at MS diagnosis; mean (SD) | | 31.76 (10.08) | 31.74 (10.07) | 33.42 (10.83) |
| MS Duration; mean (SD) | | 10.89 (5.71) | 10.86 (5.70) | 13.88 (6.20) |
| Cancer Duration | | 4.5 (2.8) | NA | 4.5 (2.8) |
| Education level; n (%) | Illiterate | 77 (0.8) | 77 (0.8) | 0 (0) |
| | Below high school | 1882 (18.8) | 1849 (18.7) | 33 (26.8) |
| | High school | 3235 (32.3) | 3187 (32.2) | 48 (39.1) |
| | Academic | 4832 (48.1) | 4790 (48.3) | 42 (34.1) |
| Sex; n (%) | Female | 7781 (77.4) | 7686 (77.4) | 95 (77.2) |
| | Male | 2268 (22.6) | 2240 (22.6) | 28 (22.8) |
| MS type; n (%) | RRMS | 8674 (86.2) | 8561 (86.3) | 103 (83.8) |
| | SPMS | 1279 (12.7) | 1261 (12.7) | 18 (14.6) |
| | PPMS | 106 (1.1) | 104 (1) | 2 (1.6) |
| DMT; n (%) | Interferon-β | 2459 (42.4) | 4208 (42.3) | 51 (41.5) |
| | Monoclonal antibodies | 2083 (20.7) | 2045 (20.6) | 38 (30.8) |
| | Dimethyl fumarate | 1110 (11.1) | 1102 (11.1) | 8 (6.5) |
| | Glatiramer acetate | 836 (8.4) | 831 (8.4) | 5 (4.1) |
| | Fingolimod | 795 (7.9) | 790 (8) | 5 (4.1) |
| | Teriflunomide | 573 (5.7) | 563 (5.7) | 10 (8.1) |
| | Other | 357 (3.5) | 356 (3.6) | 1 (0.8) |
| | None | 33 (0.3) | 28 (0.3) | 5 (4.1) |

*Abbreviation*: DMT; Disease-Modifying Therapies, MS; Multiple sclerosis, RRMS; Relapsing remitting MS, SPMS; Secondary Progressive MS, PPMS; Primary progressive MS, *NA*: *Not Assignment*

flowchart S1 Fig. Stratifying by sex, females had a rate of 123.1 (95% CI: 99.60–150.5) per 100,000 person-years, while males had a slightly higher rate of 134.9 (95% CI: 89.6–195.2). The incidence also varied by age at MS diagnosis, peaking at ages 60–64 (677.9 per 100,000 person-years) and 50–54 (353.2 per 100,000 person-years), with no reported cases in the oldest age groups (65–79). Furthermore, according to Table 2 breast cancer exhibited the highest incidence ratio at 40.17 per 100,000 person-years, followed by thyroid and digestive system cancers at rates of 18.38 and 17.36 per 100,000 person-years, respectively. Thyroid cancers showed higher rates among females (22.03 per 100,000 person-years), while digestive system bone and brain cancers had higher rates among males, at 38.54,19.27 and 19.27 per 100,000 person-years, respectively.

## People with multiple sclerosis clinical features and cancer risk

Table 3 summarizes the association between clinical features of PwMS and cancer risk. Age displayed a significant positive correlation with cancer risk in both univariate and multivariate analyses, with odds ratios (ORs) of 1.036 (95% CI: 1.021–1.051, p < 0.001) and 1.072 (95% CI: 1.045–1.100, p < 0.001), respectively. Conversely, age at MS diagnosis, sex, and MS type did not exhibit significant associations with cancer risk. Lower education levels, particularly high school and below, were linked to increased cancer risk (OR: 1.552 (95% CI: 1.016–2.370) and OR: 1.613 (95% CI: 1.001–2.600), respectively). Moreover, certain DMTs showed noteworthy associations with cancer risk. Monoclonal antibodies were associated with a substantially higher risk of cancer with an OR of 1.533 (95% CI: 1.004–2.341, p < 0.05) in the univariate

**Table 2. Incidence of cancer types among people with multiple sclerosis.**

| Type of cancer | Number | | Incidence ratio (95% CI) | | |
|---|---|---|---|---|---|
| | Overall | Female / Male | Overall | Female | Male |
| Breast | 31 | 31/0 | 40.17 (27.29–57.02) | 40.17 (27.29–57.02) | - |
| Thyroid | 18 | 17/1 | 18.38 (10.89–29.05) | 22.03 (12.83–35.27) | 4.81 (0.12–26.84) |
| Digestive organs | 17 | 9/8 | 17.36 (10.11–27.80) | 11.66 (5.33–22.14) | 38.54 (16.64–75.94) |
| Female genital organs | 11 | 11/NA | 14.25 (7.12–25.51) | 14.25 (7.12–25.51) | NA |
| Brain and other parts of central nervous system | 12 | 8/4 | 12.25 (6.33–21.41) | 10.37 (4.48–20.43) | 19.27 (5.25–49.34) |
| Bone and articular cartilage | 9 | 5/4 | 9.19 (4.20–17.44) | 6.48 (2.10–15.21) | 19.27 (5.25–49.34) |
| Skin | 7 | 5/2 | 7.15 (2.84–14.73) | 6.48 (2.10–15.21) | 9.63 (1.16–34.80) |
| Urinary tract | 6 | 4/2 | 6.12 (2.24–13.33) | 5.18 (1.41–13.27) | 9.63 (1.16–34.80) |
| Secondary and unspecified sites | 3 | 3/0 | 3.06 (0.63–8.95) | 3.88 (0.80–11.36) | - |
| Bronchus and lung | 3 | 1/2 | 3.06 (0.63–8.95) | 1.29 (0.03–7.22) | 9.63 (1.16–34.80) |
| Connective and soft tissue | 2 | 1/1 | 2.04 (0.24–7.37) | 1.29 (0.03–7.22) | 4.81 (0.12–26.84) |
| Prostate | 4 | NA/4 | 0.19 (0.05–0.49) | NA | 0.19 (0.05–0.49) |

NA: Not Assignment

model and 1.542 (95% CI: 1.009–2.357, p < 0.05) in the multivariate model, while other DMTs like Dimethyl fumarate, Fingolimod, and Glatiramer Acetate showed no significant associations. Patients not receiving DMTs demonstrated a notably elevated cancer risk compared to those receiving Interferon-β.

## Clinical features of multiple sclerosis-cancer patients

S2 Table presents a descriptive analysis of the clinical characteristics of MS-cancer patients, categorized by survival status. Deceased patients tended to be slightly younger at the time of

**Table 3. Risk of cancer in patients with MS.**

| Measurements | | OR (95% CI) | p | OR (95% CI) | p |
|---|---|---|---|---|---|
| Age | | 1.036 (1.021–1.051) | <0.001 | 1.020 (01.002–1.038) | 0.029 |
| Age at MS diagnosis | | 1.016 (0.999–1.033) | 0.064 | | |
| MS Duration | | 1.069 (1.045–1.093) | <0.001 | 1.055 (1.028–1.082) | <0.001 |
| Sex (Ref. = female) | | 1.011 (0.662–1.545) | 0.959 | | |
| MS type (Ref. = RRMS) | SPMS | 1.184 (0.714–1.963) | 0.513 | | |
| | PPMS | 1.595 (0.388–6.553) | 0.517 | | |
| Education level (Ref. = Academic) | Illiterate | - | 0.997 | | |
| | Below high school | 2.035 (1.286–3.221) | 0.002 | 1.683 (1.043–2.715) | 0.033 |
| | High school | 1.718 (1.133–2.605) | 0.011 | 1.598 (1.047–2.438) | 0.030 |
| DMT (Ref. = Interferon-β) | None | 14.74 (5.47–39.68) | <0.001 | 9.090 (3.254–25.39) | <0.001 |
| | Monoclonal antibodies | 1.533 (1.004–2.341) | 0.048 | 1.467 (0.959–2.247) | 0.078 |
| | Teriflunomide | 1.466 (0.740–2.904) | 0.273 | | |
| | Dimethyl fumarate | 0.599 (0.284–1.266) | 0.179 | | |
| | Fingolimod | 0.522 (0.208–1.313) | 0.167 | | |
| | Glatiramer acetate | 0.497 (0.198–1.248) | 0.136 | | |
| | Other | - | 0.994 | | |

Abbreviation: DMT; Disease-Modifying Therapies, MS; Multiple sclerosis, RRMS; Relapsing remitting MS, SPMS; Secondary Progressive MS, PPMS; Primary progressive MS, HR; Hazard ratio

cancer diagnosis (mean age: 39.40 years) compared to survivors (mean age: 41.14 years), and they had shorter survival time post-cancer diagnosis (mean: 2.08 years) compared to survivors (mean: 5.62 years). Notably, deceased patients had longer durations of MS before cancer diagnosis (mean duration: 8.31 years) compared to survivors (mean duration: 5.62 years).

In terms of demographics, females were predominant in both groups, accounting for 88.7% of survivors and 61.5% of deceased patients. A family history of cancer, MS, and autoimmune diseases was reported by 45.1,16.7% and 11.8% of patients, respectively.

In terms of MS treatment, a higher proportion of survivors (33.8%) received Monoclonal antibodies compared to deceased patients (26.9%). Other DMT distributions were relatively similar between the two groups.

Regarding cancer diagnosis and treatment, diagnostic methods included biopsy (100%), sonography (21.1%), and CT scans (13.8%). Digestive organs were the most common type of cancer among deceased patients (28.8%), followed by brain (17.3%) and breast cancers (15.4%). Most patients had grade 9 tumor cancer, with a higher proportion among deceased patients (73.1%) compared to survivors (56.3%). Metastasis occurred in 28.5% of cases, slightly more prevalent among living patients (29.6%). Surgical treatment was the most common approach (41.5%), followed by a combination of radiotherapy and chemotherapy (14.6%). Outcomes varied, with 42.3% of patients succumbing to cancer, 35.7% recovering, 5.7% under supervised care, and 16.3% unresponsive to the follow-up interview.

## Clinical factors and cancer mortality risk in people with multiple sclerosis

Table 4 illustrates that several factors including age, education years, MS type, age at MS onset, age at cancer onset, MS duration before cancer diagnosis, use of DMTs, and presence of

**Table 4. Risk of cancer mortality in MS patients with cancer.**

| Measurements | | HR (95% CI) | p | HR (95% CI) | p |
|---|---|---|---|---|---|
| Age | | 0.998 (0.975–1.022) | 0.868 | | |
| Education years | | 0.989 (0.922–1.061) | 0.758 | | |
| Sex (Ref. = female) | | **2.954 (1.678–5.198)** | **<0.001** | **2.683 (1.510–4.767)** | **0.001** |
| MS type (Ref. = RRMS) | SPMS | 0.819 (0.369–1.818) | 0.624 | | |
| | PPMS | - | - | | |
| MS duration | | **1.091 (1.034–1.152)** | **0.001** | **1.087 (1.029–1.148)** | **0.003** |
| Age onset MS | | 0.999 (0.974–1.024) | 0.910 | | |
| First EDSS | | 0.995 (0.391–2.533) | 0.991 | | |
| Last EDSS | | 1.201 (0.943–1.529) | 0.137 | | |
| DMT (Ref. = Interferon-β) | None | - | 0.977 | | |
| | Monoclonal | 0.696 (0.363–1.333) | 0.274 | | |
| | Teriflunomide | 0.353 (0.084–1.487) | 0.156 | | |
| | Dimethyl fumarate | 0.975 (0.340–2.797) | 0.963 | | |
| | Fingolimod | 1.328 (0.401–4.403) | 0.643 | | |
| | Glatiramer acetate | 1.340 (0.405–4.431) | 0.632 | | |
| | Azathioprine | - | 0.988 | | |
| Age-onset-Cancer | | 0.993 (0.969–1.017) | 0.544 | | |
| MS before cancer years | | 0.978 (0.932–1.026) | 0.363 | | |
| Tumor grade | | **1.134 (1.027–1.253)** | **0.013** | 1.100 (0.994–1.218) | 0.065 |
| Metastasis | | 0.980 (0.531–1.809) | 0.949 | | |

Abbreviation: MS; Multiple sclerosis, RRMS; Relapsing remitting MS, SPMS; Secondary Progressive MS, PPMS; Primary progressive MS, HR; Hazard ratio

metastasis did not exhibit significant associations with the risk of mortality from cancer. However, gender displayed a notable association, indicating that males had a significantly higher risk compared to females, with hazard ratios (HRs) of 2.954 (95% CI: 1.678–5.198, p < 0.001) and 2.683 (95% CI: 1.510–4.767, p = 0.001) in univariate and multivariate analyses, respectively. Moreover, MS duration demonstrated a significant positive association with the risk of cancer mortality, with HRs of 1.091 (95% CI: 1.034–1.152, p = 0.001) and 1.087 (95% CI: 1.029–1.148, p = 0.003) in univariate and multivariate analyses, respectively. Additionally, tumor grade exhibited a significant positive association with HRs of 1.134 (95% CI: 1.027–1.253, p = 0.013) in the univariate model, although this association was not significant in the multivariate model.

### EDSS in people with multiple sclerosis with and without cancer

Following adjustments for age, sex, type of MS, and disease duration, there was no statistically significant difference in mean EDSS scores between PwMS with cancer (1.42, SD = 1.90) and those without cancer (1.26, SD = 1.26), with a p.value of 0.766.

### Survival analysis

Fig 1 presents Kaplan-Meier survival analysis outcomes for PwMS post-cancer diagnosis, indicating notable disparities in survival rates by sex (p < 0.001). Female patients exhibited a substantially longer mean survival time of 7.204 years (S.E: 0.406), contrasting with male patients' mean survival time of 3.170 years (S.E: 0.508). Conversely, no significant disparity in survival rates was evident based on MS type (p = 0.367).

## Discussion

### Cancer incidence among people with multiple sclerosis: Higher or lower?

Understanding the prevalence of cancer among PwMS is essential for effective management and treatment, making it a subject of significant interest that has prompted numerous studies to explore this relationship.

The present study noted a substantial cancer burden among PwMS, with an overall incidence ratio of 125.6 per 100,000 person-years. Notably, this rate was lower among women at 123.1 compared to men at 134.9, although still below the global average. Globally, the incidence ratio for all cancers combined was higher in men than in women, as evidenced by age-standardized rates (ASR) of 212.5 per 100,000 for men and 186.2 per 100,000 for women [22].

A comprehensive study conducted by the INPCR provided a broad perspective into cancer incidence in the Iranian population. The overall incidence ratio of all cancers in the Iranian population was higher than in the present study, with an ASR of 158.41 per 100,000 and a crude rate of 146.22 per 100,000. Predominant cancers in men included stomach, prostate, colorectal, bladder, and lung while breast, colorectal, stomach, and thyroid were common in women corroborated with the findings of the present study. Notably, age-specific incidence ratio varied, with peaks observed at ages 75–85 in the INPCR study and at ages 60–64 and 50–54 in the current study [23].

A meta-analysis conducted in 2023 revealed that while PwMS had a decreased risk of pancreatic and ovarian cancers, they exhibited an increased risk of breast and brain cancers. Our study aligned with these findings, notably with breast cancer representing a higher proportion of all cancers at 25.2% compared to the global rate of 11.6%, and brain and nervous system cancers accounting for 9.8% in the current study, surpassing the global percentage of 1.6% [24].

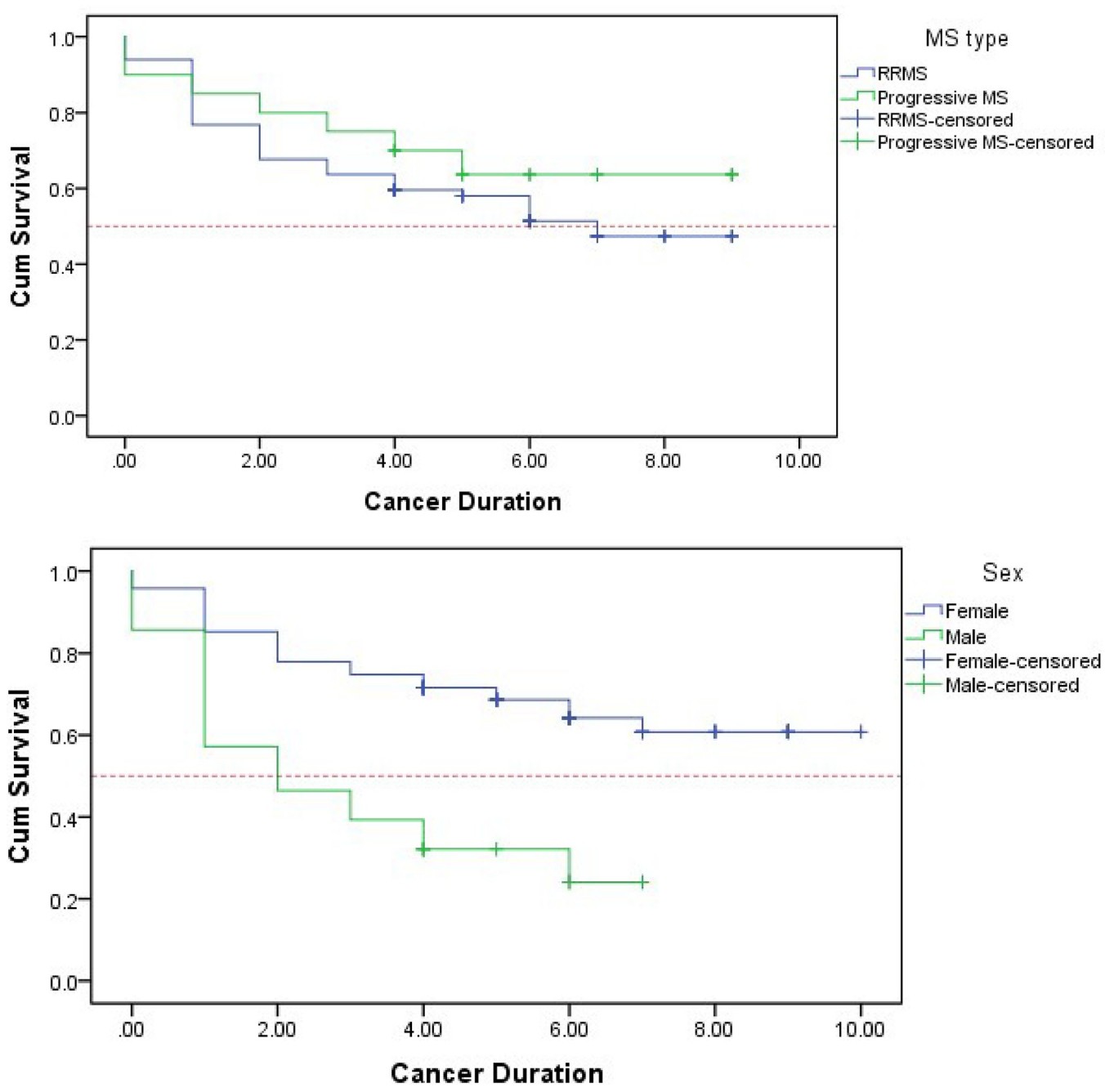

**Fig 1.**

Furthermore, two meta-analyses conducted in 2020 and 2010 exploring cancer risk in PwMS relative to the general population yielded consistent results with the current study, suggesting a lower risk of cancer among PwMS compared to controls, with pooled relative risk estimates of 0.83 and an overall odds ratio of 0.92, respectively [25, 26]. In addition, the present study found a significant positive correlation between age and cancer risk, consistent with previous findings in MS patients [27, 28].

Comparing this study with the previous cohort study conducted in Isfahan, current research, encompassing a larger PwMS cohort over an extended duration, unveiled a reduced overall cancer incidence ratio among PwMS. In contrast, the previous study, covering 1718 PwMS over 8 years, reported cancer rates similar to the general population. Moreover, while the earlier study indicated a decreased cancer risk in men with MS and an increased risk in women, our findings showed higher cancer incidence in men [29]. Furthermore, the variations in findings between the two studies may stem from differences in data collection methods, diagnostic criteria, and statistical analysis techniques.

Two recent studies have reported an increased incidence of malignancy in PwMS compared to the general population, contrasting with the findings of recent meta-analyses and our results. The Netherlands study identified breast cancer and melanoma as the most common malignancies, while the French study observed higher rates of prostate, colorectal, anal, trachea, bronchus, and lung cancers, alongside a slight increase in breast cancer among PwMS [30, 31].

The variations in cancer incidence across these studies may stem from various factors. Variances in study populations and methodologies, including differences in inclusion criteria and matching techniques, could influence observed cancer rates. Furthermore, variations in healthcare systems and access to services between countries may also be influential. Additionally, differences in environmental factors [32], genetic predispositions [33], and lifestyle behaviors [34] among populations could contribute to the differing cancer profiles.

## Cancer mortality risk in people with multiple sclerosis: Higher or lower?

The coexistence of MS and cancer poses a complex challenge in clinical management, as both conditions significantly impact mortality rates. Individuals diagnosed with MS typically experience a median decrease in life expectancy of 7.1 years, coupled with nearly a threefold rise in mortality rates compared to matched individuals from the general population. Notably, cancer ranks among the primary contributors to mortality in PwMS [35].

The present study reveals a significant positive correlation between MS duration and the risk of cancer mortality, suggesting a potential role reversal of MS in exacerbating cancer outcomes, as indicated by both univariate and multivariate analyses.

On a global scale, males generally exhibit a higher ASR of mortality from cancer (109.7) compared to females (76.8). Conversely, women tend to experience higher mortality rates from MS globally than men [22, 36]. However, the current research revealed that male MS-cancer patients demonstrated a significantly higher mortality risk and shorter survival time compared to females. This suggests that the management and treatment of cancer in PwMS may align more closely with cancer patients rather than those with MS, underscoring the need to prioritize interventions tailored to this dual condition to address similar mortality trends.

A recent study by Grytten and colleagues reported a five-fold increase in all-cause mortality and a two-fold increase in mortality following a cancer diagnosis among PwMS compared to controls [10]. Additionally, our study investigated the impact of cancer status on mortality risk, revealing that patients with higher tumor grades experienced higher mortality rates while the presence of metastases did not significantly influence mortality risk.

## Cancer diagnosis in people with multiple sclerosis

Although PwMS may not have a higher overall risk of developing cancer compared to the general population, their survival rates may be compromised. This underscores the importance of evaluating whether current MS patient care facilitates early cancer detection, which is crucial for improving survival rates. However, PwMS may face barriers to cancer screening,

potentially leading to lower rates of screen detection [37, 38]. These insights highlight the importance of establishing a comprehensive screening protocol for cancer diagnosis in PwMS.

To address this need, this study revealed several key findings regarding cancer diagnosis in PwMS, including a higher cancer diagnosis among PwMS aged 50 to 65 years old, particularly in males and those with a family history of cancer. In contrast, the age at MS diagnosis and the type of MS did not influence the likelihood of a cancer diagnosis. Notably, breast and thyroid cancers were more common among females, while digestive system and bone cancers were more prevalent among males. Furthermore, biopsy was the most common method of cancer diagnosis, followed by sonography, CT scans, and PET scans. However, these data, along with the findings of other studies, need validation and examination to develop a comprehensive guideline for the detection and management of cancer in MS patients. Currently, a French study has formulated such a guideline, which presents valuable insights but requires updating in light of new information [39].

The complexities of cancer diagnosis in PwMS are exemplified by the co-occurrence of MS and glioma, where overlapping symptoms and neuroimaging findings hinder clinical diagnosis, often resembling tumefactive demyelinating lesions on MRI [40]. New case reports continue to highlight these challenges [41].

Additionally, diagnostic challenges arise from the potential misattribution of cancer symptoms to MS, leading to longer diagnostic intervals [42]. For instance, a recent case report highlighted the misdiagnosis of advanced metastatic prostate cancer in a low-risk individual who had been mistakenly treated for MS for over 5 years due to unusual symptoms, ultimately confirmed through specialized imaging and biopsy procedures [43].

Current research revealed that deceased PwMS diagnosed with cancer were younger and had shorter survival times post-diagnosis compared to survivors. Digestive organs, brain, and breast cancers were the most frequently detected cancers among deceased patients. Additionally, the majority of deceased patients were diagnosed with significantly higher tumor grades. These findings suggest a potential delay in cancer diagnosis among PwMS, leading to more advanced cancer stages and shorter survival times. This underscores the importance of implementing improved screening protocols for cancer in PwMS, particularly at younger ages and for cancers affecting the digestive, brain, and breast regions.

### Disease-modifying therapies and cancer risk

In the field of MS treatment, the past two decades have witnessed substantial transformations owing to the introduction of numerous DMTs. These DMTs aim to alter immune responses to reduce inflammation, but their immunosuppressive effects may increase the risk of cancer in PwMS. Individuals with autoimmune conditions like MS may already have a heightened cancer risk, which could be further amplified by DMTs [8]. For instance, a recent study examined cancer incidence in Norwegian PwMS before and after the introduction of DMTs, finding no difference from 1953 to 1995 but a significant increase from 1996 to 2017, particularly in older patients, coinciding with DMT introduction [44].

The present investigation demonstrated a notable increase in cancer risk associated with monoclonal antibodies. In contrast, certain other DMTs such as Dimethyl fumarate, Fingolimod, and Glatiramer Acetate exhibited no significant associations with cancer risk. Furthermore, our findings suggested a potential protective effect of Interferon-β against cancer in PwMS, with patients receiving Interferon-β showcasing notably lower cancer risk compared to those not on DMTs.

Monoclonal antibodies have been integral in MS treatment but have raised concerns regarding increased cancer risks. For instance, Alemtuzumab, a CD52 targeting DMT, has a

history in cancer treatment, and its use in MS has been linked to various malignancies. Similarly, B cell-depleting therapies like Rituximab and Ocrelizumab, despite their efficacy in MS, have been associated with cancers. Research examining Natalizumab use in PwMS has linked it to specific malignancies such as melanoma, breast cancer, and diffuse large B-cell lymphoma. Natalizumab's mechanism of inducing susceptibility to cancer involves inhibiting T cell migration to tumor sites through α4 integrin blockade, potentially impeding antigen-specific T cell activation. In clinical trials, Ocrelizumab was linked to various malignancies, such as renal cancer, melanoma, and breast cancer [8].

Using a nested case-control design, an observational study evaluated cancer risk linked to Interferon-β treatment for MS among 5146 relapsing-onset PwMS over 48,705 person-years of follow-up, indicating no general elevated cancer risk associated with Interferon-β exposure over 12 years [45]. Additionally, the current study indicated a potential protective role against cancer among PwMS receiving Interferon-β compared to those not receiving any DMTs.

### Cancer impact on multiple sclerosis course

In the present investigation, no significant difference in mean EDSS scores between PwMS with and without cancer after adjusting for various factors. This aligns with a study investigating MS activity post-immune checkpoint inhibitor (ICI) treatment, indicating that factors like discontinuation of DMT may exacerbate MS activity more than cancer treatments themselves [46]. Similarly, a retrospective study of 43 PwMS with breast cancer showed infrequent MS relapses during treatment, with modest disability progression, consistent with oncologic outcomes in the general population [47]. Nevertheless, a separate analysis of 14 MS cases post-ICI treatment revealed rapid neurological progression in some cases, possibly due to the limited number of patients in the study, emphasizing the need for larger-scale investigations [48].

Moreover, the discovery of effective treatment strategies becomes a delicate balancing act in the presence of both MS and cancer. Therapeutic interventions must consider the potential interactions between MS treatments and cancer therapies, as well as the impact of each condition on the other's progression. Factors such as immune suppression, treatment-related toxicities, and neurologic complications necessitate personalized approaches. One critical example that urgently needs specific guidelines is women of childbearing age with both MS and cancer, especially breast cancer. Polypharmacy has been shown to be associated with clinically relevant drug–drug interactions in women of childbearing age with MS [49], and adding cancer therapy could further complicate their treatment. This highlights the challenges faced by clinicians and underscores the importance of addressing reproductive health considerations.

The present study demonstrated that surgical treatment was the most common approach, followed by a combination of radiotherapy and chemotherapy. Although the overall course of MS was not affected in cancer patients receiving these treatments in our study, further exploration of drug–drug interactions in patients with both MS and cancer is warranted.

### Strengths and limitations

This study's strengths lie in its comprehensive epidemiological scope and large sample size. These elements collectively contribute to the reliability of the findings, offering valuable insights into the subject. While this study provides valuable insights into cancer incidence among PwMS in Isfahan province, Iran, a few limitations must be considered. There is incomplete data for deceased and non-respondent patients. Furthermore, unmeasured confounders, such as lifestyle habits or genetic factors, may influence our results despite statistical adjustments.

## Conclusion

In conclusion, this study in Isfahan province, Iran, highlights the intricate relationship between MS and cancer, revealing lower incidence ratio with gender-specific and age-related variations among PwMS, underscoring the complexity of this association in terms of incidence, clinical features, and mortality risk.

Deceased patients often had higher cancer stages and shorter survival time post-cancer diagnosis, suggesting potential delays in diagnosis and the urgent need for improved screening protocols, particularly for certain cancers among younger ages.

Moreover, a significant positive correlation between MS duration and cancer mortality risk hints at a potential reversal of MS's role in exacerbating cancer outcomes.

Furthermore, cancer did not exacerbate the MS course, as evidenced by the lack of significant differences in mean EDSS scores between PwMS with and without cancer.

Additionally, the study found that monoclonal antibodies were associated with an elevated risk of cancer, highlighting the need for additional research. This underscores the importance of personalized treatment strategies, taking into account demographic factors and the choice of disease-modifying therapies linked to cancer risk. Further research is warranted to validate and expand upon these findings, which hold significant implications for clinical practice and guideline development in managing MS-cancer patients.

## Supporting information

**S1 Fig. Flow diagram of the study process and participant selection.**
(PDF)

**S1 Table. Cancer incidence among people with multiple sclerosis.**
(DOCX)

**S2 Table. Clinical characteristics of multiple sclerosis-cancer patients by survival status.**
(DOCX)

## Author Contributions

**Conceptualization:** Amirhossein Nafari, Vahid Shaygannejad, Omid Mirmosayyeb.

**Formal analysis:** Amirhossein Nafari, Saeed Vaheb, Alireza Afshari-Safavi, Omid Mirmosayyeb.

**Methodology:** Amirhossein Nafari, Saeed Vaheb, Zahra Ravankhah, Fotooheh Teimouri, Omid Mirmosayyeb.

**Supervision:** Vahid Shaygannejad.

**Writing – original draft:** Amirhossein Nafari.

**Writing – review & editing:** Saeed Vaheb, Alireza Afshari-Safavi, Zahra Ravankhah, Fotooheh Teimouri, Vahid Shaygannejad, Omid Mirmosayyeb.

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
