## [Decision Letter · Decision Letter 0]

21 Aug 2024

PONE-D-24-29820Cancer Risk and Mortality Among People with Multiple Sclerosis: A Population-Based Study in Isfahan, IranPLOS ONE

Dear Dr. Shaygannejad,

Thank you for submitting your manuscript to PLOS ONE. After careful consideration, we feel that it has merit but does not fully meet PLOS ONE’s publication criteria as it currently stands. Therefore, we invite you to submit a revised version of the manuscript that addresses the points raised during the review process.

We look forward to receiving your revised manuscript.

Kind regards,

Mohammad Reza Fattahi, M.D., M.P.H.

Academic Editor

PLOS ONE

Journal Requirements:

2. In the online submission form, you indicated that [Insert text from online submission form here]. 

7. We note that there is identifying data in the Supporting Information file <file name>. Due to the inclusion of these potentially identifying data, we have removed this file from your file inventory. Prior to sharing human research participant data, authors should consult with an ethics committee to ensure data are shared in accordance with participant consent and all applicable local laws.

-Location data

Additional Editor Comments:

Thank you for your work on this important study. The integration of MS and cancer registration systems in Isfahan and also in all Iran adds significant value to understanding the incidence and risk factors for cancer among patients with MS. However, several areas of the manuscript require clarification and revision, as detailed in the reviewer comments. Addressing these will strengthen your paper, particularly in terms of the study’s methodology, data accuracy, and the inclusion of relevant references.

I encourage you to consider the reviewers' suggestions carefully, as they will help refine your work and enhance its impact on the field.

Best regards,

Reviewers' comments:

Reviewer's Responses to Questions

**Comments to the Author**

1. Is the manuscript technically sound, and do the data support the conclusions?

Reviewer #1: Yes

Reviewer #2: Yes

Reviewer #3: Partly

2. Has the statistical analysis been performed appropriately and rigorously? 

Reviewer #1: Yes

Reviewer #2: Yes

Reviewer #3: Yes

3. Have the authors made all data underlying the findings in their manuscript fully available?

Reviewer #1: Yes

Reviewer #2: Yes

Reviewer #3: Yes

4. Is the manuscript presented in an intelligible fashion and written in standard English?

Reviewer #1: Yes

Reviewer #2: Yes

Reviewer #3: Yes

5. Review Comments to the Author

Reviewer #1: Dear Authors,

Thank you for your valuable research. I have a few minor comments to improve the quality of the study:

Abstract:

1. I recommend including the mean duration of MS diagnosis in the characteristics of patients in the Results section of the abstract.

2. Please specify which cancers are more prevalent among males and which are more prevalent among females.

Methods:

1. Could you explain why the MS National Registry of Iran was not used to study MS risks among all MS patients in Isfahan for your study? Using the registry for identifying MS patients might have provided more precise data for your study.

Results:

1. As previously mentioned, please include the mean duration of MS in the Results section under "Demographic Characteristics of People with Multiple Sclerosis."

2. Include the EDSS score and MS duration in Table 1 and change the title of Table 1 to "Demographics and Characteristics of People with Multiple Sclerosis with and without Cancer."

3. Instead of writing 'MS-only,' please write 'MS without cancer' in Table 1.

4. Include the EDSS score and MS duration in Table 2 and analyze them as the probable risk factors for cancer incidence.

5. To determine appropriate screening protocols for cancer in MS patients which is highly recommended, it is important to know the incidence of each cancer type in this population. Please add Supplementary Table 2: "Incidence of Cancer Types Among People with Multiple Sclerosis" to the main tables of the manuscript and discuss it.

Reviewer #2: This manuscript titled; Cancer Risk and Mortality Among People with Multiple Sclerosis: A Population-Based Study in Isfahan, Iran.

Many patients with multiple sclerosis face numerous challenges associated with managing a chronic illness, which may lead them to overlook early warning signs that could indicate the presence of cancer. This oversight can result in delayed cancer diagnoses, potentially leading to a higher incidence of advanced-stage cancers and increased mortality rates.

This study aims to assist neurologists and specialists in MS by highlighting the potential link between cancer and MS, and by improving patient management strategies. While the discussion provided is thorough and relevant, it lacks specific guidelines for cancer screening in this patient population. Furthermore, given the higher prevalence of women with MS who are of childbearing age, the study could offer additional insights into the challenges faced by clinicians in treating MS and addressing reproductive health considerations, especially for breast cancer among women.

Considering the noted deficiencies, the article would benefit from minor revisions to address these challenges more effectively. Additionally, the language needs refinement.

Reviewer #3: This article is well helped by the integration of MS and cancer registration systems to find the incidence rate, risk factors and mortality rate among MS patients in Isfahan.

But it has ambiguities that need to be resolved. This manuscript need Major revision for acceptance.

Below are the comments:

Title: Mortality should be written as mortality rate (in title and whole body of manuscript).

Abstract

Abstract (Background): write the goal of study before method section.

Abstract (method): please specify deputy of MUIs and other registry systems such as NMSRI.

Abstract (method): data, collected following which international protocols?

Abstract (result): insert mean age instead of average age (Also in whole body of manuscript).

Abstract (result): put per 100,000 person-years at end of sentence.

Abstract (conclusion): write 2-3 sentences summarizing the supporting points / body paragraphs and delete Further research is needed …

Introduction:

Reference 1 in nor relevant for first paragraph.

Insert relevant reference for sentence “others indicate an increased or comparable risk”.

highest MS prevalence rates for Isfahan, needs relevant reference (no reference 11).

Describe the main goal of study in last sentence.

Method:

Please insert all data sources such as National MS Registry system, if you were used in this study.

Population of Iran is more than 85milion people, you should use last national census not 2016.

Specify data collection method from PwMS in manuscript with relevant references.

What is Vice-Chancellery of MUI,?

First, you must write the full word for the first time in the text and put its abbreviation in parentheses, like central nervous system (CNS). Follow this rule throughout the whole manuscript.

You should use latest McDonald criteria not 2001 version.

Insert exact number of excluded cases for each reason such as individuals who could not be identified through their national code, those who did not participate in follow-up interviews, those with insufficient data, or those whose cancer was diagnosed prior to the onset of MS.

How many subjects were collected for interview? While you mentioned about retrospective method and collecting data from registry system, why your team had interview with subjects?

How did you calculate EDSS by interview? It is not possible. Even types of MS.

This data is not exit in MUI registry system. It is available in National MS Registry of Iran.

Reference 18 is not suitable for this method.

Result:

Average ages should be change to mean age in whole manuscript.

Incidence ratio or ratio?

The overall incidence ratio of cancer among PwMS should be calculate correctly.

oldest age groups (65-79), should be write along with.

tumor grade 9? It should be corrected.

Last paragraph of “Clinical Features of Multiple Sclerosis-Cancer Patients” is confusing.

Discussion

Discussion is too long.

Insert relevant references for last paragraph of “Cancer Incidence Among People with Multiple Sclerosis: Higher or Lower?”

No need to discuses method of cancer diagnosis.

This study's strengths are not including analysis methods, detailed consideration of various influencing factors, and high ethical and methodological standards. It is necessary to do these things.

retrospective data from specific registries may

This study's limitation is not including introduce biases and inaccuracies, limiting the generalizability of findings beyond this population, absence of a control group from the general population and Retrospective data collection risks.

Tables

The total percentage of the columns in each category must be 100%. No less and no more.

What are the column titles in tables 2 and 3?

Write all Abbreviations bellow tables such as HR.

Why, apart from the references, the statistics of other variables have not been reported?

Figure

Specify cancer duration (Year).

References

References should be update and relevant to each sentence.

You should add NMSRIs references.

The manuscript needs English editing.

6. PLOS authors have the option to publish the peer review history of their article (what does this mean?). If published, this will include your full peer review and any attached files.

Reviewer #1: No

Reviewer #2: No

Reviewer #3: No

---

## [Author Response · Author response to Decision Letter 0]

2 Oct 2024

Ms. Ref. No.: PONE-D-24-29820 

Subject: Response to Reviewers' Comments on Manuscript: PONE-D-24-29820, titled "Cancer Risk and Mortality Among People with Multiple Sclerosis: A Population-Based Study in Isfahan, Iran"

Dear Dr. Emily Chenette,

I hope this letter finds you well. we would like to express our gratitude to you and the reviewers for the valuable feedback provided on our manuscript titled "Cancer Risk and Mortality Among People with Multiple Sclerosis: A Population-Based Study in Isfahan, Iran". We appreciate the time and effort invested in reviewing our work. Following the constructive comments, we have revised the manuscript accordingly, addressing each concern raised and these alternations have been marked within the manuscript. Below is a point-by-point response to the reviewers' comments:

Comments from Reviewer 1 :

• Comment 1: [Abstract:I recommend including the mean duration of MS diagnosis in the characteristics of patients in the Results section of the abstract.] 

Response: Thank you for your valuable feedback. We have included the mean duration of MS diagnosis in the abstract as recommended.

• Comment 2: [Abstract:Please specify which cancers are more prevalent among males and which are more prevalent among females.] 

Response: Your feedback has been acknowledged. As per your suggestion, we have specified the most prevalent cancers for each gender.

“Notably, breast and thyroid cancers were more common among females, while digestive system and bone cancers were more prevalent among males.”

• Comment 3: [Methods: Could you explain why the MS National Registry of Iran was not used to study MS risks among all MS patients in Isfahan for your study? Using the registry for identifying MS patients might have provided more precise data for your study.]

Response: Our study is indeed based on data from the National Multiple Sclerosis Registry System of Iran, which we used to collect information on MS incidence among the Isfahan population. We have revised the Methods section of the manuscript to clarify this detail accordingly.

“This retrospective population-based study aimed to evaluate the incidence and prevalence of cancer among PwMS in Isfahan province, Iran, utilizing data from the National Multiple Sclerosis Registry System of Iran (NMSRI)”

• Comment 4: [Results: As previously mentioned, please include the mean duration of MS in the Results section under "Demographic Characteristics of People with Multiple Sclerosis.]

Response: We have accordingly added the mean duration of MS for People with Multiple Sclerosis with and without Cancer.

• Comment 5: [Results: Include the EDSS score and MS duration in Table 1 and change the title of Table 1 to "Demographics and Characteristics of People with Multiple Sclerosis with and without Cancer.] 

Response: Thank you for your feedback. We have added the MS duration to Table 1 and revised the table title as suggested. Unfortunately, we do not have access to the EDSS scores for all patients. The EDSS information was only obtained for patients with both MS and cancer, which was gathered through examinations by neurologists during their interviews.

• Comment 6: [Instead of writing 'MS-only,' please write 'MS without cancer' in Table 1.] 

Response: We have altered Table 1 accordingly.

Comment 7: [Include the EDSS score and MS duration in Table S2 and analyze them as the probable risk factors for cancer incidence.]

Response: We have added the MS duration and analyzed it in Table S2. However, as previously mentioned, we did not include the EDSS score since we do not have the necessary data.

Comment 8: [To determine appropriate screening protocols for cancer in MS patients which is highly recommended, it is important to know the incidence of each cancer type in this population. Please add Supplementary Table 2: "Incidence of Cancer Types Among People with Multiple Sclerosis" to the main tables of the manuscript and discuss it.]

Response: Thank you for raising this important point. We have accordingly moved Supplementary Table 2 to the main text. As you noted, we have already discussed the contents of this table and the incidence of each cancer type in the MS population in the sections "Incidence of Cancer Among People with Multiple Sclerosis" and "Cancer Incidence Among People with Multiple Sclerosis: Higher or Lower?".

Comments from Reviewer 2 :

Comment: [Many patients with multiple sclerosis face numerous challenges associated with managing a chronic illness, which may lead them to overlook early warning signs that could indicate the presence of cancer. This oversight can result in delayed cancer diagnoses, potentially leading to a higher incidence of advanced-stage cancers and increased mortality rates.

This study aims to assist neurologists and specialists in MS by highlighting the potential link between cancer and MS, and by improving patient management strategies. While the discussion provided is thorough and relevant, it lacks specific guidelines for cancer screening in this patient population. Furthermore, given the higher prevalence of women with MS who are of childbearing age, the study could offer additional insights into the challenges faced by clinicians in treating MS and addressing reproductive health considerations, especially for breast cancer among women.

Considering the noted deficiencies, the article would benefit from minor revisions to address these challenges more effectively. Additionally, the language needs refinement.]

Response: Thank you for your valuable feedback. In response to your comment, we have previously included suggestions and discussed guidelines for cancer screening in MS patients in the "Cancer Diagnosis in People with Multiple Sclerosis" section of our discussion. However, recognizing the validity of your point, we have made further revisions to this section to clarify and enhance our screening recommendations. Regarding your comment on the challenges of treating MS patients with cancer who are of childbearing age, you have raised an important point, and we are thankful for it. We have incorporated these insights at the end of the "Cancer Impact on Multiple Sclerosis Course" section of our discussion.

Comments from Reviewer 3:

Comment 1: [Title: Mortality should be written as mortality rate (in title and whole body of manuscript).]

Response: Thank you for your feedback. We have made the necessary corrections. However, as our study specifically examined mortality risk, we have chosen to use "mortality risk" instead of "mortality rate" for accuracy and consistency. 

Comment 2: [Abstract (Background): write the goal of study before method section).]

Response: Thank you for your feedback. We have revised the abstract to include the study's goal before the methods section: 

“In this study, we investigate the association between cancer risk and MS.”

Comment 3: [Abstract (method): please specify deputy of MUIs and other registry systems such as NMSRI.]

Response: Thank you for your suggestion. Data on people with multiple sclerosis (PwMS) were collected using the National Multiple Sclerosis Registry System of Iran (NMSRI)

“Data on PwMS were collected utilizing the National Multiple Sclerosis Registry System of Iran (NMSRI)”

Comment 4: [Abstract (method): data, collected following which international protocols?]

Response: Thank you for your comment. Data collection adhering to international protocols from the International Agency for Research on Cancer and the International Association of Cancer Registries. The INPCR utilized pathology reports, clinical histories, and death certificates from hospitals and pathology labs, using the International Classification of Diseases for Oncology (ICD-O-3) to code tumor characteristics. (For more information, refer to method) 

Comment 5: [Abstract (result): insert mean age instead of average age (Also in whole body of manuscript).]

Response: Thank you for your comment. We have replaced 'average age' with 'mean age' throughout the abstract and the entire manuscript.

Comment 6: [Abstract (result): put per 100,000 person-years at end of sentence.]

Response: Thank you for your suggestion. We have revised the sentence to place 'per 100,000 person-years' at the end.

Comment 7: [Abstract (conclusion): write 2-3 sentences summarizing the supporting points / body paragraphs and delete Further research is needed …]

Response: Thank you for your feedback. We have added 2-3 sentences summarizing the supporting points and removed the statement 'Further research is needed.

“Cancer incidence among 10,049 people with multiple sclerosis was significant, especially in older individuals, with breast cancer being the most common. Male gender and longer MS duration were linked to higher cancer mortality risk.”

Comment 8: [Reference 1 in nor relevant for first paragraph.]

Response: Thank you for your observation. We have revised the first paragraph and removed the reference that was deemed irrelevant.

Comment 9: [Insert relevant reference for sentence “others indicate an increased or comparable risk”.]

Response: Thank you for your suggestion. We have added more relevant references to support the sentence regarding the increased or comparable risk.

Comment 10: [highest MS prevalence rates for Isfahan, needs relevant reference (no reference 11).]

Response: Thank you for your feedback. We have updated the reference for the highest MS prevalence rates for Isfahan, ensuring that reference 11 is no longer used.

Comment 11: [Describe the main goal of study in last sentence.]

Response: Thank you for your suggestion. We have added the following sentence to the end of the paragraph: “The main goal of this study is to illuminate the cancer risks within the MS population to inform future clinical practices and research.”

Comment 12: [Please insert all data sources such as National MS Registry system, if you were used in this study.]

Response: Thank you for your feedback. We have added the following sentence to clarify our data sources: 'This retrospective population-based study aimed to evaluate the incidence and prevalence of cancer among PwMS in Isfahan province, Iran, utilizing data from the National Multiple Sclerosis Registry System of Iran (NMSRI)”

Comment 13: [Population of Iran is more than 85milion people, you should use last national census not 2016.]

Response: Thank you for your observation. We acknowledge that the Iranian population exceeds 85 million people. However, please note that the national census is conducted every ten years, and the most recent data available is from the 2016 census. We will ensure that this limitation is clearly stated in our study.

Comment 14: [Specify data collection method from PwMS in manuscript with relevant references.]

Response: Thank you for your suggestion. We have specified the data collection method in the manuscript as follows: 'Patients' information was analyzed using the National Multiple Sclerosis Registry System (NMSRI) database. All patients registered in this database have been definitively diagnosed with multiple sclerosis by a neurologist.' We will also include relevant references to support this information. (the text was completely revised)

“Data on the incidence of MS in the Isfahan population were obtained from the NMSRI.14 Detailed information regarding the data collection method for PwMS has been provided in our previous research.11 Patients diagnosed with MS in Isfahan between 1996 and 2022 were referred to the scientific committee of MUI, which consists of neurologists specializing in MS and central nervous system demyelinating disorders, for confirmation of their diagnosis using the McDonald criteria.15 Once approved, these patients were recorded in the database, and relevant information—including birth date, sex, MS onset date, course of MS, and disease-modifying therapy (DMT)—was collected for analysis.”

Comment 15: [What is Vice-Chancellery of MUI,?]

Response: Thank you for your inquiry. The Vice-Chancellery of MUI served as our contact center with the National Multiple Sclerosis Registry System (NMSRI). However, we have removed this reference from the text to enhance clarity.

Comment 16: [First, you must write the full word for the first time in the text and put its abbreviation in parentheses, like central nervous system (CNS). Follow this rule throughout the whole manuscript.]

Response: Thank you for your valuable feedback. We have revised the manuscript to ensure that all abbreviations are spelled out in full the first time they appear, followed by their abbreviations in parentheses. 

Comment 17: [You should use latest McDonald criteria not 2001 version.]

Response: Thank you for your suggestion. We have updated the manuscript to reflect the use of the latest McDonald criteria (2017).

Comment 18: [Insert exact number of excluded cases for each reason such as individuals who could not be identified through their national code, those who did not participate in follow-up interviews, those with insufficient data, or those whose cancer was diagnosed prior to the onset of MS.]

Response: Thank you for your comment., A study flowchart has been added as Figure S1 for further clarity.

Comment 19: [How many subjects were collected for interview? While you mentioned about retrospective method and collecting data from registry system, why your team had interview with subjects?]

Response: Thank you for your comment. The number of subjects collected for interviews is detailed in Figure S1, which illustrates how study data was entered. Regarding the second part of your question, some parameters of this study may not be accurately recorded in the database. Additionally, certain measures, such as the Expanded Disability Status Scale (EDSS), require neurological examinations that are conducted during the interviews. The interviews also served to verify the accuracy of the data in the database.

Comment 20: [How did you calculate EDSS by interview? It is not possible. Even types of MS.]

Response: Thank you for your comment. You are correct that the Expanded Disability Status Scale (EDSS) cannot be measured by interview alone. In this study, clinical examinations were conducted alongside patient interviews with a neurologist to accurately assess EDSS and determine the type of MS. Additionally, the type of MS is also recorded in the MS database.

“These clinical parameters were assessed through clinical examinations conducted during the interview.” 

Comment 21: [This data is not exit in MUI registry system. It is available in National MS Registry of Iran.]

Response: Thank you for your comment. This point has been corrected in accordance with your previous comments, clarifying that the data is available in the National MS Registry of Iran, rather than the MUI registry system.

Comment 22: [Reference 18 is not suitable for this method.]

Response: Thank you for your feedback. We have changed Reference 18.

Comment 23: [Average ages should be change to mean age in whole manuscript.]

Response: Thank you for your suggestion. We have updated all instances of 'average ages' to 'mean age' throughout the entire manuscript.

Comment 24: [Incidence ratio or ratio?]

Response: Thank you for your comment. We have updated the terminology from 'incidence rate' to 'incidence ratio' for clarity and accuracy.

Comment 25: [The overall incidence ratio of cancer among PwMS should be calculate correctly.]

Response: Thank you for your comment. We have recalculated the overall incidence ratio of cancer among PwMS to ensure accuracy.

Comment 26: [oldest age groups (65-79), should be write along with.]

Response: Thank you for your comment. We did not have any patients in the oldest age group (65-79), which made it impossible to calculate the incidence rate for this age range.

Comment 27: [tumor grade 9? It should be corrected.]

Response: Thank you for your observation. We have corrected the terminology regarding 'tumor grade 9'. 

“Most patients had grade 9 tumors cancer, with a higher proportion among deceased patients (73.1%) compared to survivors (56.3%).”

Comment 28: [Last paragraph of “Clinical Features of Multiple Sclerosis-Cancer Patients” is confusing.]

Response: Thank you for your feedback regarding the last paragraph of 'Clinical Features of Multiple Sclerosis-Cancer P

---

## [Decision Letter · Decision Letter 1]

11 Oct 2024

Cancer and Mortality Risks Among People with Multiple Sclerosis: A Population-Based Study in Isfahan, Iran

PONE-D-24-29820R1

Dear Dr. Shaygannejad

We’re pleased to inform you that your manuscript has been judged scientifically suitable for publication and will be formally accepted for publication once it meets all outstanding technical requirements.

Kind regards,

Mohammad Reza Fattahi, M.D., M.P.H.

Academic Editor

PLOS ONE

Additional Editor Comments (optional):

Reviewers' comments:

Reviewer's Responses to Questions

**Comments to the Author**

1. If the authors have adequately addressed your comments raised in a previous round of review and you feel that this manuscript is now acceptable for publication, you may indicate that here to bypass the “Comments to the Author” section, enter your conflict of interest statement in the “Confidential to Editor” section, and submit your "Accept" recommendation.

Reviewer #2: All comments have been addressed

Reviewer #3: All comments have been addressed

2. Is the manuscript technically sound, and do the data support the conclusions?

Reviewer #2: Yes

Reviewer #3: Yes

3. Has the statistical analysis been performed appropriately and rigorously? 

Reviewer #2: I Don't Know

Reviewer #3: Yes

4. Have the authors made all data underlying the findings in their manuscript fully available?

Reviewer #2: Yes

Reviewer #3: Yes

5. Is the manuscript presented in an intelligible fashion and written in standard English?

Reviewer #2: Yes

Reviewer #3: Yes

6. Review Comments to the Author

Reviewer #2: Comments to the Authors:

Thank you for addressing all of my comments, I appreciate the revision made.

Reviewer #3: Greeting

In my opinion, the comments have been answered appropriately and the necessary corrections have been made.

The Manuscript entitle: Cancer and Mortality Risks Among People with Multiple Sclerosis: A Population-Based Study in Isfahan, Iran could be accepted for publication in PLOS ONE, since satisfy the PLOS ONE criteria for publication.

7. PLOS authors have the option to publish the peer review history of their article (what does this mean?). If published, this will include your full peer review and any attached files.

Reviewer #2: No

Reviewer #3: No

---

## [Editor Report · Acceptance letter]

21 Oct 2024

PONE-D-24-29820R1 

PLOS ONE

Dear Dr. Shaygannejad, 

I'm pleased to inform you that your manuscript has been deemed suitable for publication in PLOS ONE. Congratulations! Your manuscript is now being handed over to our production team.

Kind regards, 

on behalf of

Dr. Mohammad Reza Fattahi 

Academic Editor

PLOS ONE